# Electrosprayed CNTs on Electrospun PVDF-Co-HFP Membrane for Robust Membrane Distillation

**DOI:** 10.3390/nano12234331

**Published:** 2022-12-06

**Authors:** Lijo Francis, Nidal Hilal

**Affiliations:** NYUAD Water Research Center, New York University, Abu Dhabi Campus, Abu Dhabi P.O. Box 129188, United Arab Emirates

**Keywords:** electrospinning, electrospraying, nanomaterials, nanostructured membrane, nanocomposites, seawater desalination, temperature polarization, flux

## Abstract

In this investigation, the electrospraying of CNTs on an electrospun PVDF-Co-HFP membrane was carried out to fabricate robust membranes for the membrane distillation (MD) process. A CNT-modified PVDF-Co-HFP membrane was heat pressed and characterized for water contact angle, liquid entry pressure (LEP), pore size distribution, tensile strength, and surface morphology. A higher water contact angle, higher liquid entry pressure (LEP), and higher tensile strength were observed in the electrosprayed CNT-coated PVDF-Co-HFP membrane than in the pristine membrane. The MD process test was conducted at varying feed temperatures using a 3.5 wt. % simulated seawater feed solution. The CNT-modified membrane showed an enhancement in the temperature polarization coefficient (TPC) and water permeation flux up to 16% and 24.6%, respectively. Field-effect scanning electron microscopy (FESEM) images of the PVDF-Co-HFP and CNT-modified membranes were observed before and after the MD process. Energy dispersive spectroscopy (EDS) confirmed the presence of inorganic salt ions deposited on the membrane surface after the DCMD process. Permeate water quality and rejection of inorganic salt ions were quantitatively analyzed using ion chromatography (IC) and inductively coupled plasma-mass spectrometry (ICP-MS). The water permeation flux during the 24-h continuous DCMD operation remained constant with a >99.8% inorganic salt rejection.

## 1. Introduction

Electrospinning and electrospraying are versatile and cost-effective techniques for fabricating nanocomposite–nanofiber membranes or substrates for a variety of applications. Electrospinning is an electrohydrodynamic process in which a polymeric droplet is subjected to a high electric potential to generate a fast jet of polymeric dope by stretching and elongation to form nanofiber mats [1,2,3,4]. In an electrospraying process, homogeneous nanomaterial dispersions are generally used instead of polymeric dope solutions. The applications of electrospinning and/or electrospraying are not limited to applications in various industry sectors in energy, healthcare, water, and the environment [5,6,7,8]. Polymeric solution or nanomaterial dispersion parameters and electrospinning/spraying process parameters are important variables to produce engineered membranes for unique applications. Solution parameters include the concentration, viscosity, surface tension, and conductivity whereas process parameters include applied potential, the distance between needle and collector, flow rate, needle diameter, and humidity [9,10]. Incorporation of nanomaterials through blending, surface modification, or an electrospraying process onto electrospun membranes enhances the membrane characteristics, such as tensile strength, hydrophobicity, antimicrobial properties, etc. [11,12,13,14,15,16,17,18,19,20]. Mitigation of fouling and rejection of humic substances in membrane-based separation processes via engineered membranes or spacers has been studied and reported by many researchers [21,22,23,24,25,26,27]. Membrane distillation (MD) is a thermally driven membrane-based separation process in which a trans-membrane vapor pressure, created due to the temperature difference on the two sides of a membrane, drives water vapors from the hot side (feed solution) to the cold side of the membrane. The majority of the MD studies are reported either on water recovery applications using newly fabricated membranes or on process engineering to enhance the efficiency of the overall MD process [28,29,30,31,32,33,34,35,36,37,38]. MD membranes can be generally fabricated by phase inversion, sintering etching, electrospinning, etc. [39]. Simulation and modeling studies have significantly supported the MD process optimization as well as a macro-level understanding of the mechanism from the lab scale to the large scale [40,41,42,43,44,45,46,47,48,49].

In 2008, Feng et al. fabricated an electrospun PVDF membrane for the MD process for the first time. The highest water permeation flux observed at a ΔT of 60 °C was 11–12 LMH in an air gap membrane distillation (AGMD) process. The state of play and recent advances of electrospun membranes for the MD process were recently reported by Francis et al. and Tijing et al. Apart from PVDF, a variety of polymeric materials, such as polyazoles, polyimides, polyurethanes, polyacrylonitriles, polysulfones, etc., have been used for MD membrane fabrication via electrospinning [5,36,37,50,51]. Various nanomaterials, such as silica, alumina, titania, graphene, silver nanoparticles, etc., have been incorporated into the MD membrane to enhance the membrane characteristics and MD process performance [52,53,54,55,56].

Electrospinning of PVDF followed by the electrospraying of alumina nanoparticles was adopted by Attia et al. for the MD membrane fabrication [13]. Shahabadi et al. demonstrated an enhanced MD performance through the fabrication of an MD membrane by the electrospinning of PVDF-Co-HFP and electrospraying of titania nanoparticles [57]. Jia et al. demonstrated the fabrication of an anti-wetting/anti-fouling MD membrane by the fabrication of a superhydrophilic/superhydrophobic membrane by the deposition of octaphenylsilsesquioxane (POSS) nanoparticles on a PVDF substrate. POSS nanoparticles enhanced the surface roughness and hydrophobicity of the MD membrane [58]. They reported that the modified membrane was superhydrophobic and showed superior water permeation flux compared to the commercial membrane. Silica nanoparticles were deposited on PVDF-HFP electrospun membranes by Su et al., and they demonstrated that the fabricated membranes were superhydrophobic and had superior anti-scaling properties [59]. On the other hand, Hong et al. reported a pore-size tunable superhydrophobic membrane for the MD process. They used polydimethylsiloxane (PDMS)/PVDF-HFP for electrospraying on an electrospun polyurethane membrane and demonstrated superior MD process performance. Their observations were validated with the aid of simulation and modeling [60]. Gethard and co-workers demonstrated the fabrication of CNT-immobilized polypropylene hollow-fiber membranes for an enhanced MD process. The CNT incorporation led to a 1.85-times increase in flux and 15-times salt rejection than those compared to the parent membrane [61]. Song and coworkers reported an electrospun membrane fabrication using a polymeric dope solution of PVDF-Co-HFP blended with CNTs and applied for salty and dyeing wastewater treatment using the direct contact MD (DCMD) process. They reported enhanced membrane characteristics and DCMD performance for the CNT-incorporated electrospun MD membrane [62].

In the current investigation, a PVDF-Co-HFP dope solution was subjected to electrospinning followed by the electrospraying of carbon nanotubes (CNTs) for the fabrication of MD membranes and applied for the DCMD desalination process using 3.5 wt. % simulated seawater as the feed solution. The CNT-modified PVDF-Co-HFP composite membrane was subjected to heat pressing before the MD process testing. Membranes were characterized for water contact angle, liquid entry pressure (LEP), pore size distribution by porometry, tensile properties by universal testing machine, surface morphology by scanning electron microscopy (SEM), and energy dispersive spectrometry (EDS). Heat pressing plays a significant role to enhance the mechanical properties of the composite membrane. Surface-modified electrospun PVDF-Co-HFP membranes with electrosprayed CNTs showed enhanced MD membrane characteristics and an increased temperature polarization coefficient (TPC) with superior desalination performance.

## 2. Materials and Methods

### 2.1. Materials

Commercially available PVDF-Co-HFP was purchased from Arkema FLUORES, Colombes, France. Dimethyl acetamide, acetone, ethanol, functionalized multiwall carbon nanotubes (MWCNT), and 1H, 1H, 2H, 2H-perfluorooctyltriethoxysilane (POTS) were purchased from Sigma Aldrich, Massachusetts, MA, USA. Sea salt was purchased from Qingdao Sea-Salt Aquarium Technology Co., Ltd. (Qingdao, China).

### 2.2. Electrospinning of PVDF-Co-HFP and Electrospraying of CNT Dispersion

The 10 wt. % PVDF-Co-HFP solution was prepared using a solvent mixture of acetone and dimethyl acetamide in a ratio of 7:3, respectively. The polymer solution was homogenized by stirring overnight at room temperature and kept for another 10 h at room temperature to escape any trapped air bubbles in the dope solution. A volume of 10 mL polymer dope was taken in a syringe and fixed on a syringe pump of an electrospinning machine (MECC NANON, Fukuoka, Japan). A 21 G syringe needle was connected to the syringe using a plastic tube. The distance between the tip of the syringe and the rotating collector drum was adjusted to 15 cm. The positive terminal of a high-voltage power supply was connected to the syringe needle, and the negative terminal was connected to the grounded rotating drum. The polymer dope flow rate, speed of the rotating drum, and speed of the spinneret were adjusted to obtain a uniform nonwoven PVDF-Co-HFP electrospun nanofiber membrane. The 0.02% functionalized MWCNT and POTS were dispersed in a 50:50 ethanol-water mixture. The ratio of MWCNT and POTS was fixed at 1:2, respectively, in an ethanol-water mixture and subjected to ultrasonication for 3 h. POTS can act as a dispersant as well as a binder for CNTs. The CNT dispersion was taken in a syringe and subjected to electrospraying on the non-woven PVDF-Co-HFP electrospun membrane using the MECC NANON electrospinning setup used for the PVDF-Co-HFP electrospinning process. The dispersion taken in the 10 mL syringe was mounted on a syringe pump, which drives the dispersion through a plastic tube to the 18 G-sized blunted syringe needle kept 15 cm above the rotating drum. The variable parameters used for the electrospinning and electrospraying processes are shown in Table 1. Figure 1 shows the schematic representation of the electrospinning and electrospraying process.

### 2.3. Heat Pressing

Pristine PVDF-Co-HFP and nanocomposite PVDF-Co-HFP-CNT membranes were subjected to the heat pressing process using a Carver Press, Auto Series Plus Hydraulic Press purchased from Mitsubishi, Japan. The membrane sheet was cut into 20 × 20 cm size, sandwiched between e-PhotoInc Heat Press Transfer Teflon sheets, and placed on the stationary bottom plate of the Carver press. The temperature and load were kept at 80 °C and 2000 N. When both plates reached the desired temperature, the heated top plate was allowed to move downwards onto the bottom stationary plate and press on the membrane sandwich at the desired force for a dwell time of 30 min.

### 2.4. Water Contact Angle Measurements

A drop-shape analyzer (DSA 100 purchased from Kruss Scientific, Hamburg, Germany) was used to measure the water contact angle of the membrane samples. The average water contact angle value measured at five different locations on each membrane sample was considered the water contact angle of the respective membranes.

### 2.5. Mean Flow Pore Size, Bubble Point Measurements, and Pore Size Distribution

The average pore size, pore size distribution, and bubble point of the PVDF-Co-HFP and PVDF-Co-HFP-CNT nanocomposite membranes before and after the heat pressing process were measured using an advanced capillary flow porometer purchased from Porous Materials Inc., (New York, NY, USA) (iPore-1500A). Circular membrane samples with a diameter of 25 mm were used for the pore size analysis. The membrane samples were kept in the sample holder of the porometer, and a Galwick fluid having a very low surface tension (15.9 dynes/cm) was dripped enough to wet the hydrophobic membrane sample. The capillary flow porometric principle is used to measure the pore size distribution in which a nontoxic liquid (Galwick fluid) is allowed to spontaneously fill the membrane pores, and an essentially non-reacting gas (N_2_) is passed through the sample to remove the Galwick liquid from the membrane pores. At lower pressures, the largest membrane pores will be emptied first and, as the pressure inserted by the nitrogen gas increases, the smallest pores get empty. The pore size distribution is obtained from the flow rate and pressure of the nitrogen gas. The pore size is inversely proportional to the pressure at which pores are empty.

### 2.6. Liquid Entry Pressure (LEP) Test

LEP is the critical pressure across a hydrophobic membrane at which the liquid starts to flow through the membrane pores [63]. In the MD process, LEP is an important parameter where the membrane has to stay non-wetted throughout the process. A bench-top LEP testing machine purchased from Convergence Minos (Convergence Industry, Drunen, The Netherlands) was used to measure the LEP of the membrane samples. Circular membrane samples with a 25 mm diameter were used for LEP measurements. Before starting the LEP test, the system automatically flushes with water to remove any air bubbles in the system. The pressure is ramped from zero in small increments (about 0.1 bar in the current investigation) until a pressure decay is observed in the system. Water starts to flow through the membrane from the point of pressure decay, and this is measured as LEP. An average value of 3 membrane samples was considered the LEP of the specific membrane.

### 2.7. Mechanical Characterizations

The mechanical properties of the PVDF-Co-HFP membranes and nanocomposite PVDF-Co-HFP--CNT membranes were characterized with a universal testing system (5965 model, Instron, Norwood, MA, USA). Standard dumbbell-shaped membrane samples were prepared using a Ray/Ran Hand Operated Cutter (RDM test equipment, Kemsing, UK) and subjected to the test using a 50 N load cell with a 2 mm/min strain rate.

### 2.8. Surface Morphology, Fiber Diameter Distribution, and EDS Images

A Thermo Fisher Field-Effect Scanning Electron Microscope (FESEM) Quanta 450, Waltham, MA, USA, was used to observe the surface morphology of the PVDF-Co-HFP and PVDF-Co-HFP-CNT membrane samples before and after the MD experiments. Gold sputter coating was employed on all membrane samples before FESEM imaging. The nanofiber diameter distribution of the electrospun membranes before and after the heat process was calculated using Image J analysis (Version 1.8). An AMETEK Octane Elect EDS detector was equipped on the FESEM machine to perform Energy-dispersive X-ray spectroscopy (EDX). Carbon sputtering was performed on the membrane samples before EDS imaging.

### 2.9. Temperature Polarization Coefficient (TPC)

Temperature polarization (TP) is a phenomenon that occurs in temperature-driven processes such as MD, in which the temperatures in the bulk feed (*T_f_*) and coolant/permeate (*T_p_*) solutions differ from the respective temperatures at the interface of the membrane and bulk solutions (feed and permeate) [64]. TP causes a reduction in the vapor pressure difference across the membrane and, thereby, the driving force of the MD process [65,66]. One of the major limitations in the MD process is the TP phenomenon. As a thermally driven separation process, heat and mass transfer are combined in the MD process, and it is very important to mitigate the TP phenomenon that occurs during the process to enhance the efficiency of the process [45,67]. The thickness of the polarization layer adjacent to the membrane surface increases as the separation process progresses, which leads to more reduction in the driving force and water vapor production. Researchers have come up with various methods to mitigate the TP phenomenon, such as employing (3D) spacers, baffles, engineered membranes, (micro) bubbling, stirring, feed flashing, employing isolation barriers, etc. [68,69,70,71,72,73,74]. TP phenomenon can be measured indirectly by using a term called TPC. TPC can be calculated using the following Equation (1) [44];
TPC = (*T_fm_* − *T_pm_*)/(*T_f_* − *T_p_*)(1)
where *T_fm_* is the feed side membrane temperature, *T_pm_* is the permeate side membrane temperature, *T_f_* is the bulk feed solution temperature, and *T_p_* is the temperature of the bulk coolant/permeate. In an ideal condition, TPC can have a maximum value of 1.

### 2.10. Direct Contact Membrane Distillation (DCMD) Experiments

A fully automated MD setup purchased from Convergence Industry, The Netherlands, was used to conduct the DCMD experiments. A DCMD membrane module with an active membrane area of 60 cm^2^ was used in all experiments. A 3.5 wt. % sea salt solution was prepared in deionized water and used as the feed solution, and tap water was used as the coolant. The DCMD experiments were conducted at feed solution temperatures of 35 °C, 40 °C, 45 °C, 50 °C, and 55 °C while keeping the coolant temperature constant at 15 °C. All the DCMD experiments were conducted at a flow rate of 60 L per hour on both sides of the membrane. A schematic representation of a direct contact membrane distillation (DCMD) experimental setup is shown in Figure 2.

Water vapor flux was calculated using the following Equation (2):*J_w_* = (*w*_2_ − *w*_1_)/*At*(2)
where ‘*J_w_*’ is the water vapor flux, ‘(*w*_2_ − *w*_1_)’ is the weight of the permeate collected at specific intervals in kilograms or liters, ‘*A*’ is the effective membrane area in square meters, and ‘*t*’ is the time in hours. Therefore, water vapor flux is represented in kg/m^2^/h or liter/m^2^/h (LMH). The temperatures of the feed and coolant at the entrance and exit of the membrane module and the conductivity of the coolant and feed solutions were measured using respective sensors periodically and recorded using a data acquisition system. Heat loss during the MD process was reduced by insulating the tubing, coolant, and feed tanks. The membrane samples were secured after 24-h continuous MD experiments for surface characterizations using FESEM and EDS. Analytical tools, such as an Inductively Coupled Plasma—Mass Spectrometer (ICP-MS Agilent 7800) purchased from Agilent Technologies, Santa Clara, CA, USA, and Ion Chromatography (IC 6000 Thermo Scientific, Waltham, MA, USA), were used for the quantitative analysis of salt ions present in the feed and permeate solutions.

## 3. Results and Discussion

### 3.1. Water Contact Angle, Heat Pressing, and Pore Size Distribution

POTS is a highly hydrophobic reagent that acts as a binder and a dispersant for CNTs in an ethanol–water mixture. Homogeneously dispersed CNTs yield an evenly distributed uniform coating on electrospun PVDF-Co-HFP membrane surfaces upon an electrohydrodynamic atomization process or electrospraying process. The observed water contact angles of the electrospun PVDF-Co-HFP and PVDF-Co-HFP-CNT membranes before and after the heat pressing process are shown in Table 2. The presence of CNTs on the PVDF membrane surface enhances the surface roughness and, thereby, the water contact angle by 3%. At the same time, the heat pressing process may cause a reduction in the surface roughness and, thereby, the average water contact angle by 3% in both the PVDF-Co-HFP membrane and CNT-modified PVDF membrane. The average flow pore size, minimum pore size, and bubble point of the electrospun PVDF-Co-HFP and composite CNT-modified PVDF-Co-HFP membranes before and after the heat pressing process are shown in Table 2.

The pore size distribution profile of pristine PVDF-Co-HFP membranes and CNT-modified membranes before and after the heat pressing process is shown in Figure 3.

It is obvious from Figure 3 and Table 2 that the minimum, average, and maximum pore sizes of the CNT-modified membranes were reduced, and pore size distribution was narrowed compared to the electrospun PVDF-Co-HFP membranes after the heat pressing process. According to Shahabadi et al., narrow membrane pore size distribution is better for enhanced MD process performance [57]. Thus, the heat pressing process helps in yielding an efficient membrane with more suitable membrane characteristics for the MD process. The average pore size of the electrospun PVDF-Co-HFP and CNT-modified PVDF-Co-HFP membranes was reduced by 36.3% and 37.5%, respectively, after the heat pressing process.

### 3.2. Liquid Entry Pressure and Mechanical Properties

Servi et al. conducted a scientific study of the effect of hydrophobicity on MD membrane wetting. They reported that the LEP increases with the water contact angle, which enhances the MD process performance [75]. The LEP of the electrospun PVDF-Co-HFP and PVDF-Co-HFP-CNT membranes was measured in the range of 120–125 KPa, whereas the LEP of the heat-pressed PVDF-HFP and PVDF-HFP-CNT membranes was measured in the range of 145–150 Kpa. The 20% increase in the LEP values in the heat-pressed membrane samples is attributed to the reduced mean flow pore sizes and narrow pore size distribution of the membranes, which is favorable to the MD process performance. Increased LEP values help retard the pore-wetting phenomenon, thereby, enhancing the membrane shelf life. The LEP values and tensile characteristics of pristine and modified membranes are given in Table 3.

Figure 4 shows the mechanical characteristics of the electrospun PVDF-Co-HFP membranes and CNT-modified membranes before and after heat pressing. The addition of CNTs on the electrospun PVDF-Co-HFP membrane surfaces enhanced the tensile strength from 40.5 KPa to 51 KPa. At the same time, elongation at the break of the CNT-modified PVDF-HFP membranes was reduced from 257% to 212%. The heat pressing process plays a significant role in enhancing the mechanical strength of both the PVDF-Co-HFP membranes and CNT-modified PVDF-Co-HFP membranes. After heat pressing, the tensile strength at break was increased from 40.5 Kpa to 89.5 Kpa for the PVDF-HFP membranes and 51.5 KPa to 90.6 KPa for the PVDF-HFP-CNT membranes. Thus, the heat pressing process helps to increase the tensile strength of the PVDF-Co-HFP membranes and PVDF-Co-HFP-CNT membranes to as high as 120% and 76%, respectively.

### 3.3. Surface Morphology

Figure 5 shows the FESEM images of the PVDF-Co-HFP membrane and PVDF-Co-HFP-CNT membrane before and after heat pressing. Figure 5 also shows the nanofiber diameter distribution of the PVDF-Co-HFP membrane before and after heat pressing. It is evident from the high-resolution microscopic images that the nanofibers are flattened, and a reduction in the surface pore size has happened upon the heat-pressing process. Thus, a slight increase from 240 nm to 285 nm in the average fiber diameter can be observed in the heat-pressed nanofibers. It is also obvious from the plots that the fiber diameter distribution has been narrowed for heat-pressed nanofiber membranes compared to the nanofiber mats before heat pressing. These observations are strengthening the narrow pore size distribution and measured pore size values of the CNT-modified electrospun PVDF-Co-HFP nanofiber membrane. CNTs distributed on the surface of the electrospun PVDF-Co-HFP membrane are also seen from the FESEM images (Figure 5c,d). CNTs on the membrane surface could create a surface roughness, and this is the reason for the increase in the water contact angle and LEP of CNT-modified PVDF-Co-HFP membranes.

### 3.4. DCMD Experiments

A series of DCMD experiments were conducted in a batch mode at various feed solution temperatures of 35 °C, 40 °C, 45 °C, 50 °C, and 55 °C while keeping the coolant temperature at 15 °C. After stabilizing the flow rate and temperatures on the feed and permeate sides, data login was initiated for the DCMD experiments. From the logged data, the water vapor flux during the DCMD test using the electrospun PVDF-Co-HFP membrane and CNT-modified electrospun-electrosprayed membrane was calculated at different feed solution temperatures. No significant flux decay was observed in each batch of the DCMD process. Salt rejection was calculated to be >99.8% in all experiments using 3.5 wt. % simulated seawater as the feed solution. Table 4 shows the percentage rejection and the results obtained from the quantitative analysis for determining the various ions present in the permeate using IC and ICP-MS analytical methods.

The calculated water vapor flux and TPC values obtained during the DCMD process while using the electrospun PVDF-Co-HFP and PVDF-Co-HFP-CNT membranes at various feed solution temperatures are shown in Figure 6. The coolant temperature was kept constant at 15 °C in all experiments. There is an exponential relationship between the trans-membrane water vapor pressure and temperature. However, the water vapor produced during the DCMD process while using the PVDF-Co-HFP and PVDF-Co-HFP-CNT membranes reached a high of 37.5 LMH and 43.4 LMH, respectively, at a feed solution temperature of 55 °C. On the other hand, at a feed solution temperature of 35 °C, the calculated flux while using the PVDF-Co-HFP and PVDF-Co-HFP-CNT membranes was found to be 6.5 LMH and 8.1 LMH, respectively. Thus, the flux enhancement while using the CNT-modified electrospun PVDF-Co-HFP via the electrospraying technique at the feed solution temperatures of 35 °C and 55 °C was calculated to be 24.6% and 15.7%, respectively. The feed and coolant temperatures at the inlets and outlets of the membrane module were recorded using data acquisition software throughout the DCMD process. TPC can have a theoretical maximum value of 1, and a high value of TPC indicates a high efficiency of the process. Mitigation of the TP phenomenon leads to a reduction in the thickness of the polarization layer adjacent to the membrane surface [76,77]. Researchers have reported different methods for creating turbulence at the membrane surface to reduce the thickness of the polarization layer at the membrane surface to increase the TPC. In the current investigation, the TPC values were calculated, and we observed that the TPC at lower feed solution temperatures was higher than that at higher temperatures. The highest TPC values while using the PVDF-Co-HFP membrane and CNT-modified PVDF-Co-HFP membrane were calculated as 0.75 and 0.87, respectively, at a feed solution temperature of 35 °C. When the feed solution temperature increased to 55 °C, the TPC values while using the aforementioned membranes were reduced to 0.71 and 0.8, respectively. Thus, enhancement in the TPC values while using the PVDF-Co-HFP membrane and CNT-modified PVDF-Co-HFP membrane at the feed solution temperatures of 35 °C and 55 °C was calculated to be 16% and 12.6%, respectively.

CNTs distributed on the electrospun PVDF-Co-HFP membrane surface via electrospraying technique act as turbulence promoters at a micro level and may reduce the thickness of the polarization layer, which would be the reason for an enhanced TPC value and water permeation flux. Increased heat and mass transfer at higher feed solution temperatures lead to faster water vapor condensation, and this may lead to an increase in the thickness of the polarization layer and, thereby, reduced TPC values.

Figure 7 shows the FESEM images of the electrospun PVDF-Co-HFP membrane and PVDF-Co-HFP-CNT membrane after a 24-h DCMD process using 3.5 wt. % simulated seawater. CNTs distributed on the electrospun PVDF-Co-HFP membranes via the electrospraying process are visible on the membrane surface even after the 24-h DCMD operation. Different salts with their respective crystal structures are deposited on the membrane surface, which is also clearly visible in the FESEM images. As discussed in the aforementioned section, the enhanced surface roughness due to the presence of CNTs on the membrane surface act as a micro-level turbulence promoter, which leads to less salt deposition and cake layer formation on the membrane surface. Prolonged salt deposition and cake layer formation reduce the water permeation flux, which may cause a pore-wetting phenomenon that leads to the salt passage through the membrane pores to the permeate side. Therefore, surface modification of MD membranes by the electrospraying of suitable nanostructured materials on the membrane surface is advantageous to enhance the membrane characteristics/efficiency and overall MD process performance. Figure 7e,f shows the EDS images taken after the 24-h DCMD experiments. It reveals the presence of sodium, potassium, magnesium, calcium, and manganese ions on the membrane surface.

Enhancements in the hydrophobicity or water contact angle LEP and mechanical strength by the incorporation of CNTs in electrospun PVDF-Co-HFP membranes are also reported by Song et al. They have also reported an increase of 30–50% MD water flux enhancement using CNT-modified membranes [62]. In this work, the CNTs were mixed with the polymeric dope solution and subjected to electrospinning for the fabrication of the MD membranes. On the other hand, in the current study, the CNTs are coated on the surface of the PVDF-Co-HFP membrane in the second step of electrospraying. This would help the nanomaterials to spread evenly throughout the membrane surface and can be available for maximum surface roughness to impart high hydrophobicity, increased water contact angle, and high LEP values. The amount of nanomaterial loading would need to be much lower in the current method of the electrospraying process compared to the already reported studies to achieve maximum membrane efficiency in terms of water permeation flux, antifouling properties, and shelf life.

## 4. Conclusions and Future Perspectives

Membrane engineering is very important in the membrane fabrication technique to produce efficient membranes in the industry. CNTs and many other nanostructured material dispersions can be efficiently electrosprayed to get evenly distributed efficient nanomaterial coating on the membrane surface. MD membrane characteristics can be tuned by an electrospray deposition technique to get desirable MD membrane properties, such as high hydrophobicity or high water contact angle (>120°), high LEP, optimum pore size (~0.2 µm), narrow pore size distribution, etc., compared to the pristine electrospun membrane. CNT modification followed by heat pressing yields mechanically robust nanocomposite membranes with improved MD membrane characteristics. A 3% increase in the water contact angle, 20% increase in the LEP, and 42.6% reduction in the mean flow pore size towards the optimum pore size were observed in the heat pressed CNT-modified electrospun PVDF-Co-HFP membranes. The tensile strength of the heat-pressed CNT-modified membrane was significantly improved by up to 120% compared to the electrospun PVDF-Co-HFP membrane. The presence of CNTs on the membrane surface before and after the MD process are visible on the FESEM images. A water vapor flux enhancement of 15.7%, 20.6%, and 24.6% was observed at a ΔT of 20 °C, 30 °C, and 40 °C, respectively. Higher TPC values and percentage water vapor flux enhancements are observed at lower feed solution temperatures because of the higher heat loss at higher feed solution temperatures compared to the lower temperatures. A 16% and 12% enhancement in the TPC values was observed at the feed solution temperatures of 35 °C and 55 °C, respectively. A >99.8% inorganic salt rejection was observed through quantitative analytical tools (IC and ICP-MS) while conducting the DCMD process using a 3.5 wt. % simulated seawater feed solution. Thus, electrohydrodynamic atomization using appropriate nanomaterial dispersion can be recommended as an efficient tool for the surface modification of MD membranes.

CNTs have shown strong antimicrobial properties. In terms of future perspectives, the antibacterial properties of CNT-coated PVDF-Co-HFP membranes have to be explored. Simulation and modeling studies would give a clear understanding of the mechanism of heat and mass transfer during the MD process while using electrospray-deposited CNT-modified PVDF-Co-HFP membranes. Modeling tools will also give an idea of turbulence patterns at the micro level during the MD process. Atomic force microscopic (AFM) studies could reveal the nature of adhesive forces between the electrospun PVDF-Co-HFP membrane and CNTs deposited through the electrospraying process.

## Figures and Tables

**Figure 1 nanomaterials-12-04331-f001:**
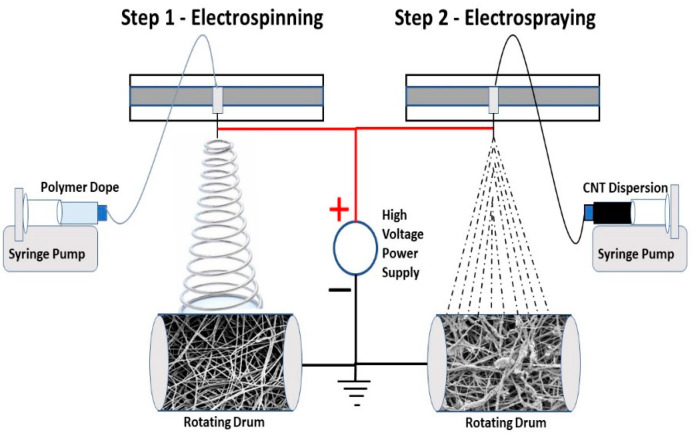
Schematic representation of nanocomposite membrane fabrication via electrospinning of PVDF-Co-HFP and electrospraying of CNT.

**Figure 2 nanomaterials-12-04331-f002:**
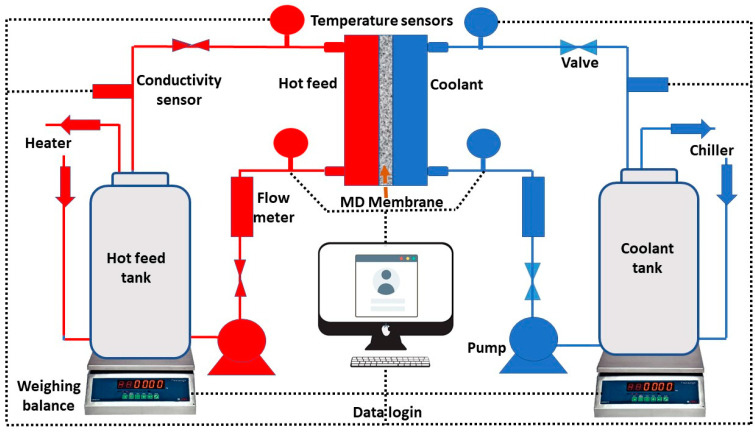
Schematic representation of experimental DCMD setup.

**Figure 3 nanomaterials-12-04331-f003:**
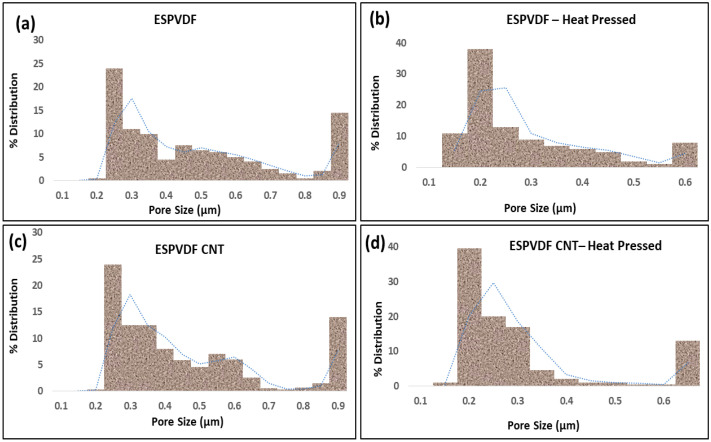
Percentage pore size distribution of the electrospun PVDF-Co-HFP and CNT-modified PVDF-Co-HFP membranes before and after heat pressing (**a**) Pristine electrospun membrane before heat pressing, (**b**) Pristine electrospun membrane after heat pressing, (**c**) CNT-coated electrospun membrane before heat pressing, and (**d**) CNT-coated electrospun membrane after heat pressing.

**Figure 4 nanomaterials-12-04331-f004:**
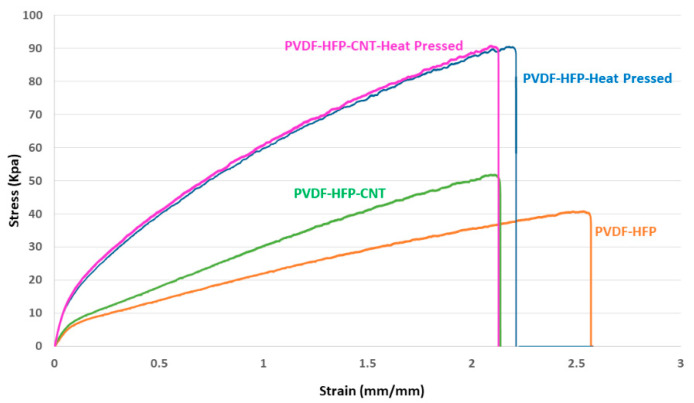
Mechanical characteristics of the electrospun PVDF-FHP and CNT-modified PVDF-HFP membranes before and after heat pressing.

**Figure 5 nanomaterials-12-04331-f005:**
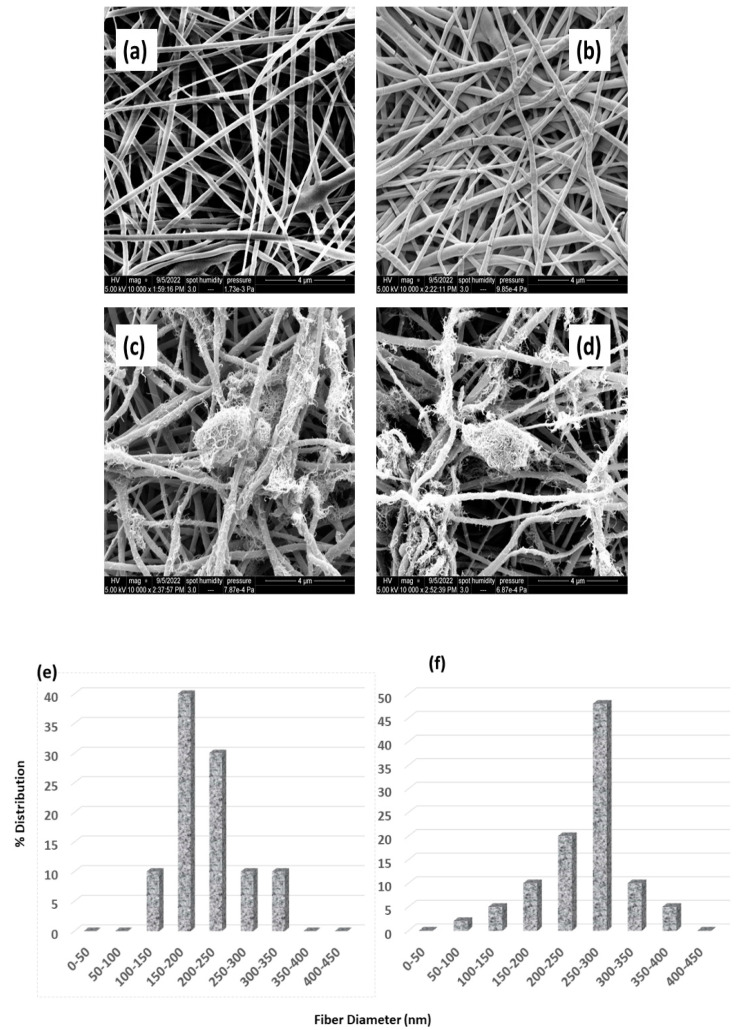
FESEM images of (**a**) PVDF-Co-HFP membrane before heat pressing, (**b**) PVDF-Co-HFP membrane after heat pressing, (**c**) PVDF-Co-HFP-CNT membrane before heat pressing, and (**d**) PVDF-Co-HFP-CNT membrane after heat pressing. (**e**) Nanofiber diameter distribution of PVDF-Co-HFP membrane before and (**f**) after heat pressing process.

**Figure 6 nanomaterials-12-04331-f006:**
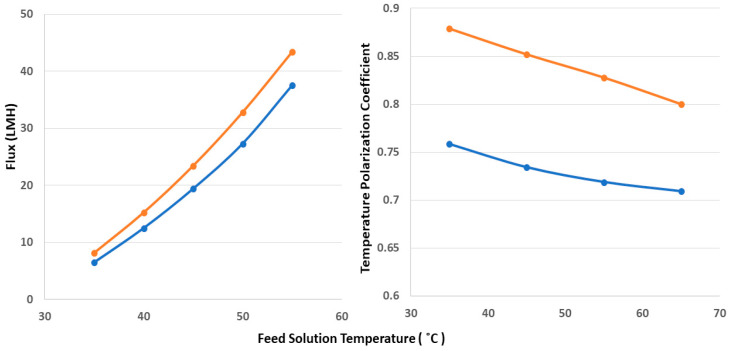
Water permeate flux and TPC values during a 24-h DCMD test using PVDF-Co-HFP (blue color) and CNT-coated PVDF-Co-HFP (orange color) membranes at a constant coolant temperature of 15 °C.

**Figure 7 nanomaterials-12-04331-f007:**
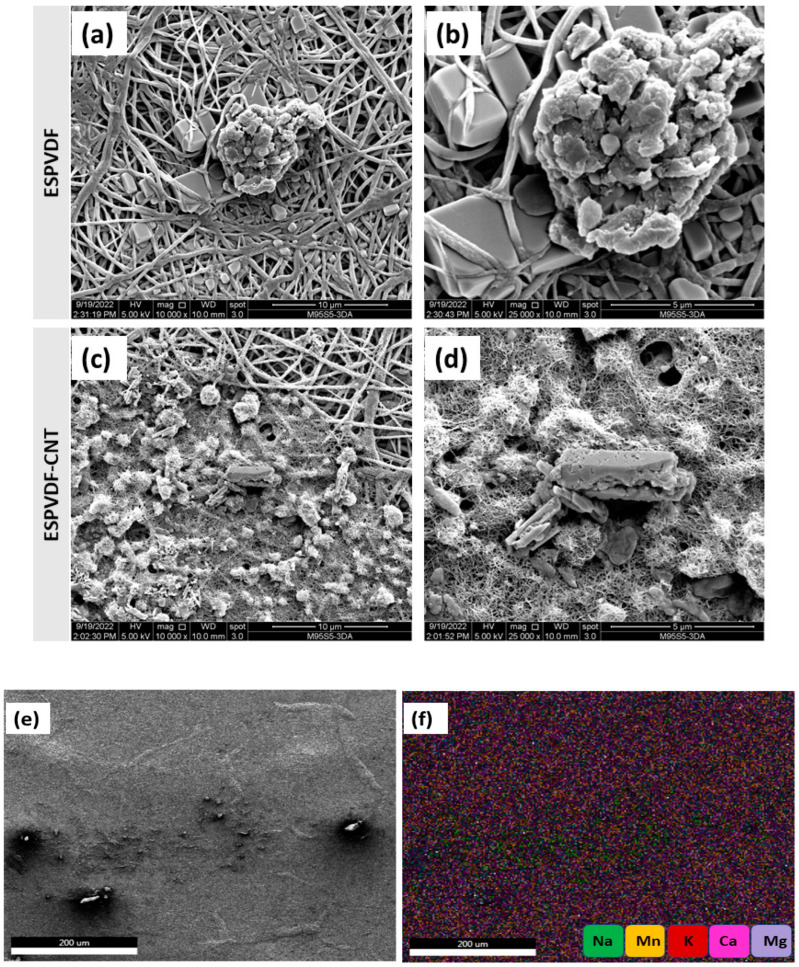
FESEM images of the electrospun PVDF-Co-HFP (**a**,**b**) and PVDF-Co-HFP-CNT (**c**,**d**) membranes after the 24-h DCMD operation, and the EDS images (**e**,**f**) of the CNT-modified PVDF-Co-HFP membrane after the MD process.

**Table 1 nanomaterials-12-04331-t001:** Electrospinning and electrospraying parameters.

Parameters	Electrospinning	Electrospraying
Solution/Dispersion Concentration	10 *w*/*w* % PVDF-Co-HFP in 7:3 Acetone:DMAc	0.02 *w*/*w* % MWCNT in 50:50 Ethanol:Water
Applied Potential (kV)	25	20
Working Distance (cm)	15	15
Flow Rate (mL/hour)	1	0.5
Speed of Collector Drum (rpm)	100	100
Spinneret Speed (mm/S)	5	5

**Table 2 nanomaterials-12-04331-t002:** Water contact angle and pore size distribution of the PVDF-Co-HFP and PVDF-Co-HFP-CNT membranes.

Membrane	ESPVDF-HFP	ESPVDF-HFP-HP	ESPVDF-HFP-CNT	ESPVDF-HFP-CNT-HP
Water contact angle	136 ± 2	132 ± 3°	140 ± 3°	136 ± 2°
Minimum pore size (µm)	0.083	0.051	0.091	0.049
Mean Flow Pore Size (µm)	0.448	0.285	0.411	0.257
Pore size at the bubble point or maximum pore size (µm)	0.921	0.615	0.92	0.715

**Table 3 nanomaterials-12-04331-t003:** LEP and tensile characteristics of the PVDF-HFP and PVDF-HFP-CNT membranes before and after heat pressing.

Membrane Characteristics	PVDF-HFP	PVDF-HFP-CNT	PVDF-HFP Heat Pressed	PVDF-HFP-CNT Heat Pressed
LEP (KPa)	120	125	145	150
Elongation at break (%)	257	212	221	212
Tensile Strength (KPa)	40.5	51.5	89.5	90.6

**Table 4 nanomaterials-12-04331-t004:** Amount of inorganic salt ions present in the permeate from the DCMD process and the percentage rejection.

Salt Ions	Permeate (ppm)	Rejection (%)
Sodium	24.80	99.8
Magnesium	2.69	99.8
Potassium	1.48	99.6
Calcium	1.51	99.7
Lithium	<0.0	100
Boron	0.037	99.6
Chloride	46.38	99.8
Sulfate	5.49	99.8
Nitrate	0.99	99.8
Bromide	0.0094	99.9

## Data Availability

Data presented in this article will be available upon request.

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
