# Peer review of "Electrosprayed CNTs on Electrospun PVDF-Co-HFP Membrane for Robust Membrane Distillation"

_nanomaterials, 2022, doi:10.3390/nano12234331_

Round 1
Reviewer 1 Report
The paper is well-planned and well-written, however, some improvements need to be provided before the acceptance:
1. DCMD - line 19 - unexplained abbreviation (I guess the authors meant Direct Contact MD).
2. The presented study generally sounds moderately novel, however, it would be worth mentioning that NCTs in water desalination applications (also MD) were investigated in other works (e.g. https://doi.org/10.1021/am100981s). In the paper https://www.nature.com/articles/srep41562 the electrospun PVDF-HFP membranes with embedded NCT as MD were presented! The authors should provide some mentions of similar works in this field of science from the past and try to explain better what is the element of novelty in the presented manuscript. For instance: I would like to know if is it better to deposit NCT on the fiber's surface or to put this inside the fiber.
Author Response
Comments and Suggestions for Authors, and point-by-point answers to each comment.
The paper is well-planned and well-written; however, some improvements need to be provided before the acceptance:
- DCMD - line 19 - unexplained abbreviation (I guess the authors meant Direct Contact MD).
Ans: Corrected. We have gone through the whole manuscript to check for typos and other errors.
- The presented study generally sounds moderately novel, however, it would be worth mentioning that NCTs in water desalination applications (also MD) were investigated in other works (e.g. https://doi.org/10.1021/am100981s). In the paper https://www.nature.com/articles/srep41562 the electrospun PVDF-HFP membranes with embedded NCT as MD were presented! The authors should provide some mentions of similar works in this field of science from the past and try to explain better what is the element of novelty in the presented manuscript. For instance: I would like to know if is it better to deposit NCT on the fiber's surface or to put this inside the fiber.
Ans: The given reference and another reference related to CNT-modified PVDF MD membrane have been newly cited and discussed in the manuscript as shown below.
Lines 81-88
“Gethard and co-workers demonstrated the fabrication of CNT immobilized polypropylene hollow fiber membrane for enhanced MD process. The CNT incorporation led to 1.85 times increase in flux and 15 times salt rejection than those compared to the parent membrane [61]. Song and coworkers reported the electrospun membrane fabrication using a polymeric dope solution of PVDF-Co-HFP blended with CNTs and applied for salty and dyeing wastewater treatment using the direct contact MD (DCMD) process. They have reported enhanced membrane characteristics and DCMD performance for the CNT-incorporated electrospun MD membrane [62].”
Lines 388-400.
“Enhancements in the hydrophobicity or water contact angle LEP and mechanical strength by the incorporation of CNTs in electrospun PVDF-Co-HFP membrane are also reported by Song et al. They have also reported an increase of 30-50% MD water flux enhancement using CNT-modified membrane [62]. In this work, CNTs were mixed with the polymeric dope solution and subjected to electrospinning for the fabrication of MD membranes. On the other hand, in the current study, CNTs are coated on the surface of the PVDF-Co-HFP membrane in the second step of electrospraying. This would help the nanomaterials to spread evenly throughout the membrane surface and can be available for maximum surface roughness to impart high hydrophobicity, increased water contact angle, and high LEP values. The amount of nanomaterial loading would need much lower in the current method of electrospraying process compared to the already reported studies to achieve maximum membrane efficiency in terms of water permeation flux, antifouling properties, and shelf life.”
Reviewer 2 Report
In this paper, Lijo coworkers investigate the electrospraying of CNTs on an electrospun PVDF-Co-HFP membrane to fabricate robust membranes for Membrane Distillation (MD) process. The CNT modification followed by heat pressing yields mechanically robust nanocomposite membranes with improved MD membrane characteristics. This work is of importance for practical distillation applications and this electrohydrodynamic atomization using appropriate nanomaterial dispersion can be recommended as an efficient tool for the surface modification of MD membranes. However, in my opinion, there are still some minor issues that need to be clarified. So I recommend a minor modification before publication. The followings are some point-by-point questions and comments on this manuscript:
1. For the pore size of ESPVDF and ESPVDF-CNT (Figure 3), it seems that there are two different dominant pore sizes of the membrances of 0.25μm, 0.9μm (before heat pressing), and 0.2μm, 0.6μm (after heat pressing). From the point of structure, why are there two dominant pore sizes of the composited membrances?
2. In the morphology analysis, (Figure 5), we can see that after adding CNTs, there will be large particles on the surface of the film. Are these agglomerated CNTs? How are most of the CNTs combined on the surface, are they across the surface of PVDF, or are they combined in other ways
3. In Figure 7, we can see that after the DCMD process in different salts, there are many crystals deposited on the membrance surface. So I’m curious that how is the durability and recyclability of the PVDF-Co-HFP/CNT composite membrance?
4. The format of all references is very inconsistent, please carefully revise the full-text including references according to the submission guideline. And there are also some typos, e.g. P8 Line 265. Please carefully refine the text.
Author Response
Comments and Suggestions for Authors, and point-by-point answers to each comment.
In this paper, Lijo coworkers investigate the electrospraying of CNTs on an electrospun PVDF-Co-HFP membrane to fabricate robust membranes for Membrane Distillation (MD) process. The CNT modification followed by heat pressing yields mechanically robust nanocomposite membranes with improved MD membrane characteristics. This work is of importance for practical distillation applications and this electrohydrodynamic atomization using appropriate nanomaterial dispersion can be recommended as an efficient tool for the surface modification of MD membranes. However, in my opinion, there are still some minor issues that need to be clarified. So I recommend a minor modification before publication. The followings are some point-by-point questions and comments on this manuscript:
- For the pore size of ESPVDF and ESPVDF-CNT (Figure 3), it seems that there are two different dominant pore sizes of the membrances of 0.25μm, 0.9μm (before heat pressing), and 0.2μm, 0.6μm (after heat pressing). From the point of structure, why are there two dominant pore sizes of the composited membrances?
Ans: This is very common in the electrospinning process. Since the membranes are non-woven mats and the pore sizes can be distributed in a wide range. However, the most important is that after heat pressing we get non-woven membranes with appropriate pore sizes and narrow pore size distribution for the MD process.
- In the morphology analysis, (Figure 5), we can see that after adding CNTs, there will be large particles on the surface of the film. Are these agglomerated CNTs? How are most of the CNTs combined on the surface, are they across the surface of PVDF, or are they combined in other ways.
Ans: Yes. CNTs are agglomerated but uniformly throughout the membrane surface, which enhances the surface roughness and thereby the water contact angle and LEP. This is advantageous for MD membranes for enhanced process performance.
- In Figure 7, we can see that after the DCMD process in different salts, there are many crystals deposited on the membrance surface. So I’m curious that how is the durability and recyclability of the PVDF-Co-HFP/CNT composite membrance?
Ans: In membrane-based separation processes it is a very common and unavoidable experience to have membrane fouling and scaling. But it can be mitigated or the propensity of fouling and scaling can be reduced to an extent by the surface modification of membranes using antifouling or anti-scaling additives as we have used CNTs in the current investigation. In our past experiences, we have also used silver nanoparticles and mesoporous organo-silica nanoparticles for fabricating fouling-resistant membranes. Reverse Osmosis membranes are commercialized and have been used in the Seawater Desalination industry for the past several decades and their lifetime is about 5-7 years. With periodic and efficient backwashing/cleaning cycles, MD membranes are also supposed to have a similar lifetime as RO membranes. Some of the MD pilot plants already have several years of history of continuous operation.
10.1016/j.memsci.2005.04.049
- The format of all references is very inconsistent, please carefully revise the full-text including references according to the submission guideline. And there are also some typos, e.g. P8 Line 265. Please carefully refine the text.
Ans: All the references are automatically extracted from Mendeley Software. However, we have double-checked all the references to make them consistent. We have gone through the whole manuscript to check for typos and other errors.
A couple more references related to CNT-modified MD membranes have been newly cited and discussed in the “Introduction” and “Discussion” sections of the manuscript as shown below.
Lines 81-88
“Gethard and co-workers demonstrated the fabrication of CNT immobilized polypropylene hollow fiber membrane for enhanced MD process. The CNT incorporation led to a 1.85 times increase in flux and 15 times salt rejection than those compared to the parent membrane [61]. Song and coworkers reported the electrospun membrane fabrication using a polymeric dope solution of PVDF-Co-HFP blended with CNTs and applied for salty and dyeing wastewater treatment using the direct contact MD (DCMD) process. They have reported enhanced membrane characteristics and DCMD performance for the CNT-incorporated electrospun MD membrane [62].”
Lines 388-400.
“Enhancements in the hydrophobicity or water contact angle LEP and mechanical strength by the incorporation of CNTs in electrospun PVDF-Co-HFP membrane are also reported by Song et al. They have also reported an increase of 30-50% MD water flux enhancement using CNT-modified membrane [62]. In this work, CNTs were mixed with the polymeric dope solution and subjected to electrospinning for the fabrication of MD membranes. On the other hand, in the current study, CNTs are coated on the surface of the PVDF-Co-HFP membrane in the second step of electrospraying. This would help the nanomaterials to spread evenly throughout the membrane surface and can be available for maximum surface roughness to impart high hydrophobicity, increased water contact angle, and high LEP values. The amount of nanomaterial loading would need much lower in the current method of electrospraying process compared to the already reported studies to achieve maximum membrane efficiency in terms of water permeation flux, antifouling properties, and shelf life.”